# Renal Microcirculation Injury as the Main Cause of Ischemic Acute Kidney Injury Development

**DOI:** 10.3390/biology12020327

**Published:** 2023-02-17

**Authors:** Ewa Kwiatkowska, Sebastian Kwiatkowski, Violetta Dziedziejko, Izabela Tomasiewicz, Leszek Domański

**Affiliations:** 1Department of Nephrology, Transplantology and Internal Medicine, Pomeranian Medical University in Szczecin, Powstańców Wlkp, 72, 70-204 Szczecin, Poland; 2Department of Obstetrician and Gynecology, Pomeranian Medical University in Szczecin, Powstańców Wlkp, 72, 70-204 Szczecin, Poland; 3Department of Biochemistry and Medical Chemistry, Pomeranian Medical University in Szczecin, Powstańców Wlkp, 72, 70-204 Szczecin, Poland

**Keywords:** kidney microcirculation, acute kidney injury, renin–angiotensin–aldosterone system

## Abstract

**Simple Summary:**

Acute kidney injury can result from multiple factors. The main cause is reduced renal perfusion. Kidneys are susceptible to ischemia due to the anatomy of microcirculation that wraps around the renal tubules. In the kidney, cortical and medullary superficial tubules have a large share in transport and require the supply of oxygen for energy production, while it is the cortex that receives almost 100% of the blood flowing through the kidneys and the medulla only accounts for 5–10% of it. This difference makes the tubules present in the superficial layer of the medulla very susceptible to ischemia. Impaired blood flow causes damage to the inner layer of vessels and thrombosis. The next stage is the disintegration of these vessels. The phenomenon of destruction of small vessels is called peritubular rarefaction, attributed as the main cause of further irreversible changes in the damaged kidney leading to the development of chronic kidney disease. In this article, we will present the characteristic structure of renal microcirculation, its regulation, and the mechanism of damage in acute ischemia, and we will try to find methods of prevention with particular emphasis on the inhibition of the renin–angiotensin–aldosterone system.

**Abstract:**

Acute kidney injury (AKI) can result from multiple factors. The main cause is reduced renal perfusion. Kidneys are susceptible to ischemia due to the anatomy of microcirculation that wraps around the renal tubules–peritubular capillary (PTC) network. Cortical and medullary superficial tubules have a large share in transport and require the supply of oxygen for ATP production, while it is the cortex that receives almost 100% of the blood flowing through the kidneys and the medulla only accounts for 5–10% of it. This difference makes the tubules present in the superficial layer of the medulla very susceptible to ischemia. Impaired blood flow causes damage to the endothelium, with an increase in its prothrombotic and pro-adhesive properties. This causes congestion in the microcirculation of the renal medulla. The next stage is the migration of pericytes with the disintegration of these vessels. The phenomenon of destruction of small vessels is called peritubular rarefaction, attributed as the main cause of further irreversible changes in the damaged kidney leading to the development of chronic kidney disease. In this article, we will present the characteristic structure of renal microcirculation, its regulation, and the mechanism of damage in acute ischemia, and we will try to find methods of prevention with particular emphasis on the inhibition of the renin–angiotensin–aldosterone system.

## 1. Introduction

Acute kidney injury (AKI) can result from multiple factors. The main cause is reduced renal perfusion [1]. AKI is associated with longer and more costly hospitalization, an increased risk of chronic renal failure, and a higher mortality rate [2]. Twenty percent of hospitalized patients develop AKI [2,3]. In developed countries, approximately 2 million people die each year due to AKI-related complications [4,5]. A reduction in blood flow is the common pathway into the pathophysiology of ischemic AKI. The causes of this disturbance can be different: Decreased intravascular volume, hypotension, drug-induced ischemia, and sepsis. Rather than alterations in whole-organ perfusion, it may be regional changes in renal blood flow that explain the loss of kidney function. The outer medulla, for instance, has been shown to be particularly vulnerable to ischemia. The mechanism of ischemic injury in this area can be explained by the anatomy of the capillary network microcirculation. The capillary network in this area is poor, and even in healthy individuals, the outer medulla is poorly oxygenated. The proximal straight tubule PST-S3 fragment present in this area cannot switch to anaerobic glycolytic metabolism, which is why it needs sufficient oxygen supply for ATP production to maintain active transport. Impaired flow in the microcirculation of the renal medulla causes damage to these vessels, which results in the initial damage to the endothelium, with an increase in its prothrombotic and pro-adhesive properties. This causes congestion in the microcirculation of the renal medulla. The next stage is the migration of pericytes, which causes the complete disintegration of these vessels. Pericytes transform into cells causing fibrosis. The phenomenon of destruction of small vessels is called peritubular rarefaction, the main cause of further irreversible changes in the damaged kidney leading to the development of chronic kidney disease. In this article, we will present the characteristic structure of renal microcirculation, its regulation, the mechanism of damage in acute ischemia, and we will try to find methods of prevention with particular emphasis on the inhibition of the renin–angiotensin–aldosterone system.

## 2. The Anatomy of Renal Microcirculation

Renal circulation supplies blood to every glomerulus in the nephron. The nephron, together with the blood vessels, is referred to as the nephrovascular unit. As mentioned earlier, 25% of the blood volume flows through the kidneys, although this much blood only passes through the cortex to maintain glomerular filtration [6]. The renal artery divides into interlobar, arcuate, and interlobular arteries, the latter of which gives rise to the afferent glomerular arterioles. In a healthy kidney, blood first flows through the glomerulus and then exits through the efferent arteriole. The efferent arteriole gives rise to the so-called the postglomerular circulation that wraps around the renal tubules to form the peritubular capillary (PTC) network. The efferent arterioles running from the cortical glomeruli vascularize the cortical tubules, while the glomeruli present close to the medulla—known as juxtamedullary glomeruli—give rise to the efferent arterioles responsible for blood supply to the medullary tubules—its name is juxtamedullary circulation [7,8]. The superficial nephrons have short efferent arterioles that form a capillary network primarily around the proximal and distal convoluted tubules. The efferent arterioles of the glomeruli located close to the medulla are longer and wider, and form vasa recta (VR)—straight descending arterioles (or descending vasa recta—DVR) and straight ascending venules (or ascending vasa recta—AVR), which give rise to the dense capillary plexus penetrating the medullary interstitium. The straight vessels run a similar route to the loop of Henle, while the direction of the flow is identical to the flow of urine in the loop of Henle. This system is responsible for blood supply to the proximal straight tubule (PST) and the medullary thick ascending limb (mTAL) of the loop of Henle. Both the cortical proximal and distal tubules, as well as the PST and the ascending limb of the loop of Henle, are considerably involved in tubular transport. The proximal straight tubule is the S3 segment of the proximal tubule. The tubular segments in the deep part of the medulla—the collecting duct and the thin descending and thin ascending limbs of the loop of Henle—have only a slight share in tubular transport and can hence draw their energy from anaerobic metabolism. Although both the compartments—cortical and medullary superficial—have a large share in the transport and require the supply of oxygen for ATP production, it is the cortex that receives 100% of the blood flowing through the kidneys, while the medulla only accounts for 5–10% of it. This discrepancy results from their anatomy—blood to the medulla only flows from the efferent arterioles of the juxtamedullary glomeruli (a small number). Additionally, the angle at which the arterioles exit the glomeruli is acute. This looks very different in the cortex, with plentiful glomeruli and efferent arterioles and a mild exit angle. It is for this reason that the S3 fragment, namely, the PST, is considered the most sensitive to ischemia. Importantly, this problem also affects the mTAL fragment [9]. While the PST cannot produce ATP from anaerobic glycolysis, the mTAL is capable of that, although on a small scale that prevents active transport from being maintained [9,10] (Figure 1).

PTC—peritubular capillary, EA—efferent arteriole, AA—afferent arteriole, PCT—proximal convoluted tubule, DCT—distal convoluted tubule, CD—collecting duct, DVR—descending vasa recta, AVR—ascending vasa recta, mTAL—medullary thick ascending limb, PST—proximal straight tubule, JMG—juxtamedullary glomeruli.

## 3. Outer Medulla of the Kidney—A Region Most Vulnerable to Ischemia

Investigations of ischemic AKI pathophysiology have shown that the pathologies behind it are diverse: Tubular damage, inflammation, and vascular damage. Many researchers claim that the part that is most exposed to ischemia is not the cortex but rather the outer portion of the medulla [11]. As early as 20 years ago, multiple studies on animal models proved that it is primarily the tubules—the PCT and the mTAL—that are damaged during ischemia [12,13,14]. Parts of the PST and the mTAL are located within the cortex, which is why there is no visible clinical sign of injury, which only manifests through disturbed sodium and potassium reabsorption as exemplified by the reduced activities of Na-K ATPase and the NaH exchanger. The partial pressure of oxygen also varies between the cortex and the medulla: The former is approximately 50 mmHg and the latter is approximately 10–20 mmHg. The core can be claimed to be exposed to hypoxia in its physiological state [15]. A slight drop in renal perfusion may cause medullary injury. The medulla defends itself against reduced blood supply. Less oxygen, and therefore fewer ATPs, reduces NaK-ATPase activity, which prevents the operation of the sodium-potassium-chloride cotransporter, thus decreasing Na, K, and Cl reabsorption in the mTAL. A higher sodium concentration in the collecting duct provides an osmotic stimulus to the macula densa, which releases ATP/adenosine and paracrine vasopressin that constricts the afferent arteriole, thus reducing perfusion through both the glomeruli and glomerular filtration, thereby reducing the Na load flowing to the macula densa. Reduced glomerular filtration, meaning a reduced Na load, leads to lower oxygen consumption in the medulla. This phenomenon is referred to as tubuloglomerular feedback (TGF). The mechanism by which macula densa constricts the afferent arterioles is not fully understood. In experiments on animal models administered with furosemide, tissue partial pressure of oxygen in the medulla increased—Na-K-Cl cotransporter inhibition reduced Na/K-ATPase activity in the basal part of the tubular epithelium, which reduced oxygen consumption [16,17,18].

In their very well-conducted animal model experiment, Zhang et al. proved that the first to suffer injury in hypoxia is medullary capillary circulation. The injury occurred when there was no inflammatory infiltration, no damage to the tubules, no reduced glomerular flow, no glomerular capillary obliteration, and no glomerulosclerosis. It manifested through a decrease in the density of the tubular capillaries. It was only when examined at a later stage after ischemia that the kidneys revealed a typical histopathological picture that is observed in AKI, i.e., inflammatory infiltration in the parenchyma, glomerular lesions, and fibrosis. Another rapidly emerging change in this experiment was the increased expression of hypoxia-inducible factor HIF-1α, an element that appears in hypoxia [19]. This experiment showed that during ischemia, the first part of the kidney to be injured was the medullary capillaries. Any other changes were a consequence of this disorder.

In an attempt to examine the medullary hypoxia proposal, one researcher produced systemic hypoxia in rats and failed to observe reduced filtration or fractional excretion of sodium [20,21]. In a second group of animals, the same author caused renal ischemia by closing renal arteries for 45 min, which caused renal function impairment after 24 h [20]. This experimental model showed that it was not hypoxia but the impaired flow that was the main factor causing damage to the peritubular capillaries in the medulla. Other studies have shown that the real weakness of the kidneys, or more specifically the medulla, is circulatory congestion during reduced kidney flow. It is characterized by erythrocytes aggregating in the outer part of the medulla in a process referred to as vascular congestion due to hypoperfusion. This has been observed both in human and animal kidneys. More importantly, even after renal perfusion is restored, congestion is maintained in the outer section of the medulla, causing a lack of reperfusion in the peritubular capillaries [22,23,24]. Congestion has been observed to be maintained for 24 h after reperfusion [22]. More recently, this phenomenon has been studied in patients with circulatory insufficiency. The main prognostic factor for the development of cardiorenal syndrome in heart failure has been found to be venous congestion in the renal medulla, rather than reduced renal flow [9,25]. Attempts to reduce hematocrit or increase volemia through the supply of mannitol have mitigated this effect [22,26]. In hypoperfusion, erythrocyte aggregation in the peritubular capillaries is the primary driver of the lesions, hence the importance of hemodilution and hydration in patients with renal hypoperfusion. This phenomenon is a consequence of anatomical factors. The peritubular capillary network is very branched, which results in a naturally slow flow and difficulty mobilizing aggregated erythrocytes. Another explanation is that the swollen tubular epithelial cells suck water out of the surrounding parenchyma. Parenchyma also has a network of capillaries that are thus deprived of water. These theories have been tested in an experimental animal model—rats [22,26]. Blood congestion is observed in all areas of the kidney, but it goes away immediately after reperfusion, only being maintained in the medulla. The cortex receives a larger volume of blood flow with a higher perfusion pressure and a higher velocity. Therefore, after reperfusion, the cortical flow becomes normal in an instant. Medullary circulatory congestion occurs even after a mere reduction, and not necessarily an arrest, of renal perfusion, as a rat model experiment suggests [27]. In his experiment, Owji et al. clamped the renal artery in one group and the renal vein in a second group for 30 min and found that the most profound damage was to the tubular epithelium in the PST and the mTAL after the closure of the renal vein and that the damage occurred not in the cortex but in the corticomedullary region. This experiment showed that the most extensive injury takes place due to blood congestion in the renal medulla [28]. The question arises as to why evolution has allowed for such poor microcirculation to develop in the renal medulla. The answer is that this is all due to urine concentration and so-called countercurrent multiplication. In the urine concentration process, the gradient of osmolality increasing from the outer part of the medulla to the renal papilla is of key importance. Osmotically active compounds (NaCl and urea) move from the AVR to the medullary interstitium and then to the DVR. This system retains these compounds in the medullary interstitium through recycling between the AVR and the DVR to maintain the countercurrent exchange. The high medullary osmolality facilitates urine concentration in the loop of Henle, which has variable permeability for water and other substances [29]. A slow flow through the straight vessels is crucial to maintain this high gradient, hence the kidney’s exposure to ischemia [10]. In summary, the renal medulla is exposed to ischemic damage due to the tubules’ high oxygen demand, the reduced blood supply of the PST and mTAL resulting from the anatomy of the nephrovascular unit, and the superficial part of the medulla being prone to persistent hyperemia as a consequence of ischemia.

## 4. Regulation of Renal Circulation-Microcirculation

Renal circulation is highly capable of adapting to systemic pressure (a MAP between 80 and 180 mmHg) without changing the glomerular filtration rate. Cortical and medullary blood flow and distribution are regulated by way of the paracrine system—nitric oxide (NO) and other vasoactive substances, the myogenic mechanisms, tubuloglomerular feedback (TGF), connecting tubule glomerular feedback (CTGF), the renin–angiotensin–aldosterone (RAAS) system, and the sympathetic nervous system (SNS) [30,31,32]. Afferent arteriole resistance has a major impact on glomerular perfusion regulation. Intraglomerular pressure and glomerular filtration are regulated by afferent and efferent arteriole tension. Despite the arterial pressure changes during the day, these arterioles maintain constant pressure in the renal glomerulus [33]. Their tension is regulated by vasodilators such as NO and prostaglandin E2, or by vasoconstrictive factors such as endothelin, angiotensin II, and adenosine [33]. The afferent arteriole contains mechanoreceptors, which cause the arteriole to constrict where systemic pressure is high and dilate where it is low. This is referred to as ‘myogenic regulation’. On the one hand, it maintains glomerular flow at low pressure, while on the other, it protects the glomerulus and postglomerular circulation from damage due to high pressure [32]. The TGF and CTGF systems operate in opposition. A high sodium level in the distal tubule entering the macula densa causes the afferent arteriole to constrict, while high sodium in the collecting duct causes the afferent arteriole to dilate. Tubuloglomerular feedback (TGF) is an important intrarenal regulatory mechanism, which acts to stabilize renal blood flow, GFR, and the tubular flow rate. The main part of this negative feedback system is the Juxtaglomerular Apparatus (JGA). This is located between the thick ascending limb of TAL and the vascular pole of the glomerulus. The JGA consists of the macula densa, the mesangial cells, and the afferent arteriole, the main effector site for the TGF. A higher sodium concentration in the collecting duct provides an osmotic stimulus to the macula densa, which releases ATP/adenosine and paracrine vasopressin that constricts the afferent arteriole, thus reducing perfusion through both the glomeruli and glomerular filtration, thereby reducing the Na load flowing to the macula densa. Reduced glomerular filtration, meaning a reduced Na load, leads to lower oxygen consumption in the medulla [16,17,18]. The RAAS is activated by the macula densa in the event of a low sodium level in the distal tubule, low arterial pressure, and sympathetic system activation, which activates the juxtaglomerular cells inducing them to secrete renin. It converts angiotensinogen into angiotensin I and then into its active form—angiotensin II—using a converting enzyme. Angiotensinogen is supplied to the kidney from the outside (as produced in the liver). It is also produced locally by proximal tubular cells. Angiotensin II causes constriction of both the afferent and efferent arterioles. Angiotensin II stimulates vasopressin release by the pituitary gland and aldosterone secretion by the adrenal cortex. In the kidney, vasopressin work via receptors V1 and V2. V1 receptors are found on vascular smooth muscle. They are coupled through phospholipase C, their activation produces vasoconstriction via the elevation of intracellular calcium. V2 receptors mediate the antidiuretic effect by adenylyl cyclase, the activation of protein kinase A, and the insertion of water channels (aquaporins) into the luminal membranes of renal collecting duct cells to absorb water [33]. The medullary microcirculation is particularly sensitive to the vasoconstrictor effects of the vasopressin works by the V1 receptor. Studies on rats by Cowley et al. have shown that renal medullary interstitial infusion of selective V1 agonists can reduce medullary blood flow by 20–40%. Cowley et al. showed that V2 receptor stimulation in the presence of a V1 receptor antagonist increased blood flow to this region. They proved the opposite effect of vasopressin on renal microcirculation depending on the stimulated V1 or V2 receptors. Vasopressin stimulates the release of nitric oxide (NO) via the stimulation of V2 receptors on the medullary-collecting duct. NO have the opposite effect to the vasoconstrictive effect of activation receptor V1. In the case of prolonged action of vasopressin, the effect of increased NO synthesis in the outer medulla dominates. These changes are essential to maintain blood flow through the medulla [34,35]. In a study on rats by Edwards et al. vasopressin at physiological concentrations caused contraction of the efferent arteriole of the glomerulus. This action increases glomerular filtration, but additionally reduces renal microcirculation [36]. In patients with septic shock, Wang et al. administered terlipressin and performed renal contrast-enhanced ultrasound (CEUS). In the group treated with terlipressin, they found a better flow in the renal cortical microvasculature than in the group without the drug [37]. A post hoc analysis of the VASST trial “ addition of terlipressin to norepinephrine in septic shock and effect of renal perfusion: A pilot study” using creatinine-based Risk, Injury, Failure, Loss, End-stage renal disease (RIFLE) criteria demonstrated that in patients with the RIFLE category ‘Risk’, vasopressin was associated with a decrease in mortality, a decrease in progression to RIFLE ‘Injury’ and ‘Failure’, a decrease in creatinine, and a decrease in the need for renal replacement therapy. In human clinical trials, the effect of vasopressin in septic shock is beneficial to renal circulation. Due to the different results of different experiments, further research is necessary [38,39].

The SNS is activated as a response to a signal from the carotid artery and aortic arch baroreceptors. Other factors activating this system are hypoxia and the activation of chemoreceptors in the carotid arteries. The sympathetic system constricts all renal vessels. Sympathetic innervation has been found in the glomerular arteries, the vasa recta, the macula densa, and the renal tubules [33,40,41,42]. The endothelium lining of the renal vessels plays an important role in regulating the homeostasis of microcirculation as an endocrine organ [43]. The endothelium produces NO, endothelin-1, adrenomedullin, adenosine, and cyclooxygenase metabolites, i.e., thromboxane and prostacyclin. Their imbalance causes flow disturbances in the microcirculation [44,45]. Endothelin is one of the strongest vasoconstrictors in the kidney. Its production in the kidney, especially the medulla, is many times higher than it is in other organs [46,47]. Endothelin has a very strong constrictive effect on the glomerular arteries and on the descending vasa recta, thus having the capacity to exacerbate medullary damage [47]. The endothelial surface is covered by the glycocalyx composed of proteoglycans and glycosaminoglycans. It forms a coating that protects the structure and functioning of the endothelium and controls the tightness of the vessels preventing macromolecules from escaping. It also has an effect on the endothelium’s interaction with leukocytes. The kidney contains arteriovenous fistulae. In the cortex, they bypass microcirculation and are a protective mechanism against excessive oxygen loads, thus preventing oxidative stress [47,48]. In the medulla, they connect the VRD to the VRA, which appears to be necessary to maintain the operation of the urine condensation mechanism [49].

## 5. Microcirculation in AKI

What happens in the event of hypovolemia and/or hypotonia-hypoperfusion? Firstly, the RAAS is activated, which causes the afferent and efferent arterioles to constrict. This reduces glomerular filtration and sodium flow to the PST and mTAL tubules, thus decreasing their consumption of ATP, but unfortunately, glomerular hypoperfusion entails microcirculatory hypoperfusion, especially affecting the medulla. Secondly, the sympathetic nervous system is activated, which not only causes the afferent arteriole but also medullary microcirculation (vasa recta) to constrict. Thirdly, in response to hypovolemia, the myogenic mechanism constricts the afferent arteriole. Moreover, arteriovenous fistulae may open in the renal medulla and cortex, exacerbating hypoxia. In an animal model, it was demonstrated that despite renal hypoperfusion and hypoxia, high oxygen pressure in the renal vein was maintained—which could indicate that fistulae were being opened under these pathological circumstances. [18] Microcirculatory ischemia causes damage to the glycocalyx layer on the endothelial surface, endothelial damage, abnormal reactions to vasoactive substances, increased permeability, leukocyte migration causing inflammation, and the production of reactive oxygen species causing microcirculatory failure and increased hypoxia. A detailed description of the damage mechanism can be found in the next chapter. A patient hospitalized in this condition receives fluids, often with a high sodium content. With a constricted efferent arteriole, this will result in increased filtration of water and salt and their elevated supply to the medulla, boosting oxygen consumption in the PST and the mTAl and elevating the amount of sodium delivered to the collecting duct. This activates the TGF and increases the contraction of the afferent arteriole. It appears that while applying fluid resuscitation, which is the first step in treating ischemic AKI, it is important to increase the flow in medullary microcirculation, although the improvement of macrocirculation is not always associated with the improvement of microcirculation. Researchers have observed a lack of correlation between macrocirculation and microcirculation during hypoperfusion [50,51]. Some papers have shown that fluid resuscitation itself is insufficient to restore renal perfusion with cortical and medullary oxygenation [52,53,54]. In a study on an animal model, Lima et al. concluded that the restoration of blood flow through renal microcirculation, together with systemic MAP compensation, failed to reverse hypoperfusion in the microcirculation, in which congestion and blocked vessels were identified [55]. The previously mentioned AKI animal model experiment carried out by Zahn et al. showed that the first lesions to develop in ischemia took the form of damage to medullary microcirculation and its rarefaction [19]. In his work, Chvojka et al. proved that blood congestion in microcirculation is a major early-ischemia-related disorder. He did not notice any other lesions developing in the kidney. He stressed that a well-preserved renal flow—macrocirculation—does not guarantee microcirculatory flow [56]. Another investigator also confirmed the decrease in the partial pressure of oxygen in the renal medulla during experimentally induced sepsis with renal hypotension and hypoperfusion [57]. A growing number of publications have been highlighting that renal microcirculation is the most important aspect of ischemic AKI prevention and treatment.

## 6. Peritubular Capillary Rarefaction as a Consequence of Acute Ischemia and a Cause of Progression to Chronic Kidney Disease

AKI may follow various courses, from full recovery to progression toward CKD. In a meta-analysis of 13 cohort studies, CKD and end-stage internal failure (ESRF) were found in 25.8 persons/100 patients who had had AKI [58,59]. The risk ratio for developing CKD is 8.8 and for ESRF it is 3.1 in persons with a history of AKI [58,59]. The exact mechanism behind acute injury transforming into chronic damage is not known. Investigators have implicated many different explanations. One of the proposals is referred to as peritubular capillary rarefaction, where the capillary density is reduced. As mentioned above, some researchers believe that this is the first lesion to develop in ischemic AKI. It seems that this lesion, as a primary one, exacerbates hypoxia and predisposes the kidney to the progression into and development of CKD. PTC rarefaction has been described in diabetic nephropathy, hypertensive nephropathy, advanced IgA nephropathy, congenital nephrotic syndromes, lupus nephritis, and allograft nephropathy, suggesting that PTC loss is common in CKD [60,61,62,63,64,65,66,67,68,69]. Renal cells have a high regenerative potential, but this does not apply to proximal tubular and renal capillary cells [30]. In their inner layers, the cortical peritubular capillaries and the straight ascending venule (AVR) capillaries have endothelial cells (ECs), which have large pores 60–80 nm in diameter allowing the easy movement of molecules and water. These pores are spanned by diaphragms made of fibers and glycosaminoglycans [29,70,71,72]. The ECs are covered by the aforementioned glycocalyx. On the outside, these vessels are lined by pericytes. The straight descending arteriole (DVR) capillaries have a different design, with no pores, and are lined by smooth muscle cells [32]. The DVR acts as both a transport vessel (a capillary) and a resistance vessel (an arteriole). Some of the disturbances typical of a developing PTC rarefaction do not apply to the DVR. In many studies on mouse models, PTC density correlated with the GFR better than the degree of fibrosis did. The atrophy of PTCs and straight vessels exiting the efferent arteriole affects the GFR. It has been proven in a rat model that tubular atrophy causes the GFR to fall [73]. The main mechanism leading to PTC rarefaction is damage to these vessels’ endothelial cells by reducing angiogenic factor expression. In response to renal injury/ischemia, endothelial cells initially proliferate and then die through apoptosis [69]. The initial proliferation is associated with increased vascular endothelial growth factor VEGF expression. However, the next step involves a reduction of the expression of VEGF and its receptor. This causes endothelial cell apoptosis. [30,74]. The expression of another angiogenic factor—angiopoietin-1 (Angpt-1)—is also significantly reduced [64]. Anti-angiogenic factors such as thrombospondin-1 and Angpt-2 are stimulated [30,71,72,75]. The damage causes an infiltration by macrophages, which secrete inflammatory cytokines—Il1B, IL-6, and TNF-alpha—which counter VEGF activity [76,77,78]. The initial lesions to the capillaries also consist of endothelial cell thickening and losing pores. These changes occur very quickly [79,80]. Endothelial cell thickening indicates that the cells have been activated. When activated, they secrete heparanases and hyaluronidases that destroy glycocalyx. Without its glycocalyx, the capillary endothelium becomes procoagulant and proadhesive [81]. This disturbance may explain why the medullary capillaries experience congestion and erythrocyte aggregation, which does not go away after reperfusion. One medical experiment concluded that the GFR correlated negatively with the degree of damage to glycocalyx [81]. For a reason that is yet unknown, the renal capillary endothelium in an animal model and in humans has a very low proliferative potential compared to the vessels in other organs. Only 0.5–1% of the cells proliferate [82]. In the cortical and AVR capillaries, the endothelial cells are covered by a layer of pericytes responsible for the synthesis of the basement membrane and the maintenance of the structure of these vessels [81]. In injury/ischemia, pericytes have been observed to migrate from vessels causing their disintegration [83]. During their migration into the parenchyma or the tubules, they change their profibrotic potential to become fibroblasts and myofibroblasts and initiate fibrosis. This phenomenon also speeds up endothelial damage [83]. Two animal model studies of toxin-induced acute kidney injury showed renal capillary rarefaction on the fourteenth day of the toxic agent’s action [84,85]. A rat unilateral ureteral obstruction model showed PTC rarefaction as well [74]. An animal ischemia-reperfusion model also revealed medullary capillary rarefaction as early as week 4 [85]. Similarly, models with induced glomerulonephritis also confirmed the rarefaction of these vessels [1,63,86,87]. Ehling et al. created three mouse models of renal injury—ischemia-reperfusion, obstructive, and Alport syndrome. Using micro-computed tomography imaging, they noticed gradual renal capillary rarefaction. In repeated studies, they confirmed that microcirculatory changes preceded the development of fibrosis [80]. Microcirculation disorder within the renal medulla initiates a series of events that not only damage the peritubular vessels but also result in tubular atrophy and fibrosis of the parenchyma. These changes are irreversible and are the main cause of the progression of acute renal failure to chronic renal failure over a longer period of time. Figure 2 shows the damage to peritubullar capillaries and the formation of the peritubular rarefaction phenomenon.

In response to renal ischemia, endothelial cells (ECs) die through apoptosis. The causes of ECs apoptosis are the reduction of expression of angiogenic factors—the vascular endothelial growth factor (VEGF) and angiopoietin-1 (Angpt-1)—and the elevation of anti-angiogenic factors such as thrombospondin-1 and Angpt-2. ECs secrete heparanases and hyaluronidases that destroy glycocalyx, and the endothelium becomes procoagulant and proadhesive. This disturbance may explain why the medullary capillaries experience congestion and erythrocyte aggregation, which does not go away after reperfusion. Then pericytes migrate from vessels causing their disintegration. During their migration into the parenchyma or the tubules, they change their profibrotic potential to become fibroblasts and myofibroblasts and initiate fibrosis.

## 7. Therapy for Medullary Hypoperfusion

As mentioned above, the renin–angiotensin–aldosterone system is activated in the event of hypotension, hypoxia, and sympathetic system activation. All three stimuli are present in ischemic AKI. The most potent compound within the RAAS system is angiotensin II (AngII). As discussed above, AngII is created both in the systemic circulation and in the kidney itself, with its concentration in the kidney being multiple times higher. All elements of the RAAS are found within the kidney, the system can therefore be produced locally and exert an autocrine effect [58,88]. It reduces glomerular flow and the glomerular filtration rate, but most importantly, it decreases the volume of blood exiting the glomerulus through the efferent arteriole, thus curbing the cortical and medullary tubular capillary flows. Efferent arteriole constriction increases glomerular pressure and hyperfiltration, promoting sclerosis [89]. Blocking the RAAS during AKI can reduce the risk of PTC rarefaction with subsequent progression to CKD. In the event of hypovolemia and hypotension, the use of drugs that inhibit the RAAS system will exacerbate these disorders. Therefore, it seems best to start using these drugs after normalizing the blood pressure and volume. Excessive intrarenal RAAS activity has been observed to occur in AKI. This activity was studied by measuring the urine concentration of angiotensinogen. Its concentration correlated with the severity of AKI and was a predictor of the development of AKI [89,90]. Multiple studies have demonstrated an upregulating effect of angiotensin II on the expression levels of various agents adversely impacting the progression of chronic lesions. It promotes parenchymal fibrosis by inducing fibroblast proliferation through the upregulated expression of transforming growth factor-beta (TGF-β), connective tissue growth factor, fibronectin, and type 1 collagen [91,92,93]. AngII causes the extracellular matrix to accumulate by activating plasminogen activator inhibitor-1 (PAI-1) and the tissue inhibitor of matrix metalloproteinase-1 (TIMP1) that counters metalloproteinase activity [94,95,96] AngII activates the proinflammatory transcription factor nuclear factor kappa-light-chain-enhancer of activated B cells (NF-κB) [97]. AngII also upregulates the expression of vascular cellular adhesion molecule-1, intra-cellular adhesion molecule-1, integrin, and chemokines such as monocyte chemoattractant protein-1, and activates T cells. All of these exacerbate the inflammatory infiltration of the glomerulus and the renal parenchyma [98,99]. A study performed by Wu et al. found that losartan (an AngII receptor inhibitor) administered to rats subjected to hypoxia-restricted capillary rarefaction in the renal cortex, damage to the renal tubules and HIF-1α expression, and reduced angiotensin II concentration in the renal cortex and in the plasma [100]. Another study on rats proved that shortly after reperfusion, the concentration of AngII increased, with prolonged maintenance of the receptor-bound AngII in the glomerulus—also 120 h after reperfusion. It has also been demonstrated that in rats administered with losartan at the time of reperfusion, renal function was quick to go back to normal (as measured according to the creatinine level) [101]. In their study on rats, Zhang et al. concluded that the administration of ACEi or an AngII receptor blocker prior to kidney injury induced by the excision of five-sixths of the kidney prevented the occurrence of PTC rarefaction compared to the control group [102]. Zhang claimed that the lower PTC rarefaction involved downregulated expression of HIF-1α. This implied that the inclusion of angiotensin II blockers reduced renal ischemia [102]. A large study involving patients undergoing coronary artery bypass surgery using extracorporeal circulation showed that AKI developed less frequently in those who had received a preoperative administration of the angiotensin-treatment enzyme inhibitor (ACEi) [103]. Another clinical study involving patients undergoing cardiac surgeries showed that the administration of ACEi reduced the postoperative risk of developing CKD [104] Similar results were achieved by Roberts et al., who, within 30 days of cardiac surgery, included an ACEi inhibitor, thus reducing AKI incidence [105]. In 40% of the 96,983 patients hospitalized due to AKI (stages II and III), a RAAS inhibitor was included upon discharge. During a year-long follow-up, no recurrence of AKI was found in either the study or control groups. A higher mortality rate was found in untreated patients. As the follow-up lasted for one year only, it was not long enough to be able to assess CKD progression. However, it showed that it is worthwhile to include RAAS inhibitors at least for a short time after the AKI episode [106]. Bidulka et al., as well as Brar et al., came to similar conclusions [107,108]. In hemodynamic disorders, we often tend to discontinue ACEi and AngII receptor inhibitor administration to reduce the risk of AKI developing. However, judging by the above-described mechanisms, we should not do so. One large meta-analysis showed that the discontinuance of RAAS inhibitors prior to cardiac surgeries or coronary catheterization failed to reduce AKI incidence [109,110]. Another large meta-analysis by Cheng et al. showed that the inclusion of RAAS inhibitors after the onset of AKI (most after recovery, but in one study, during AKI treatment) resulted in a lower risk of AKI recurrence and development of CKD [111]. Hypovolemia and hypotension are connected to the vasoconstriction of precapillaries. Vasopressor agents, despite the arterial pressure rise, frequently do not overcome precapillary vascular resistance. We call this phenomenon the vascular bottleneck. There are clinical trials that have tried to combine vasopressors and vasodilalatators to overcome this phenomenon and improve circulation, especially in septic shock. Dobutamine, nitroglycerin, and prostacyclin analogs were used as dilatators [112].

## 8. Conclusions

The inclusion of ACEi or AngII receptor inhibitors in AKI is often associated with a certain degree of resistance because efferent arteriole dilation reduces the glomerular filtration rate. It needs to be remembered, however, that failure to administer these agents will increase renal medullary ischemia. If one does not intend to administer them in the acute stage of the condition, they should include them quickly in order to counter the progression of chronic lesions. Long-term follow-up of AKI patients is necessary to be able to assess how many will develop CKD in the long term. Clinical trials with the inclusion of RAAS blockers for AKI prevention and CKD development prevention are also required. The undeniable difficulty is that AKI has very different underlying pathophysiologies.

## Figures and Tables

**Figure 1 biology-12-00327-f001:**
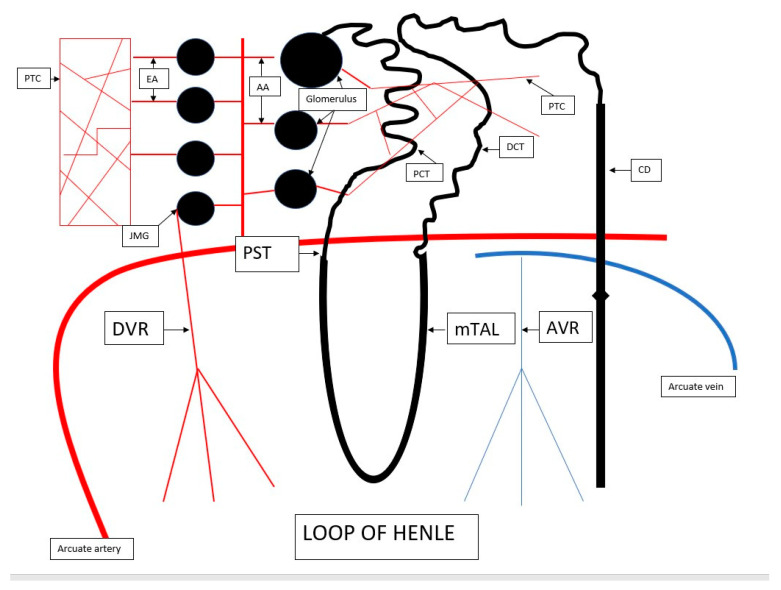
Renal microcirculation anatomy—the nephrovascular unit.

**Figure 2 biology-12-00327-f002:**
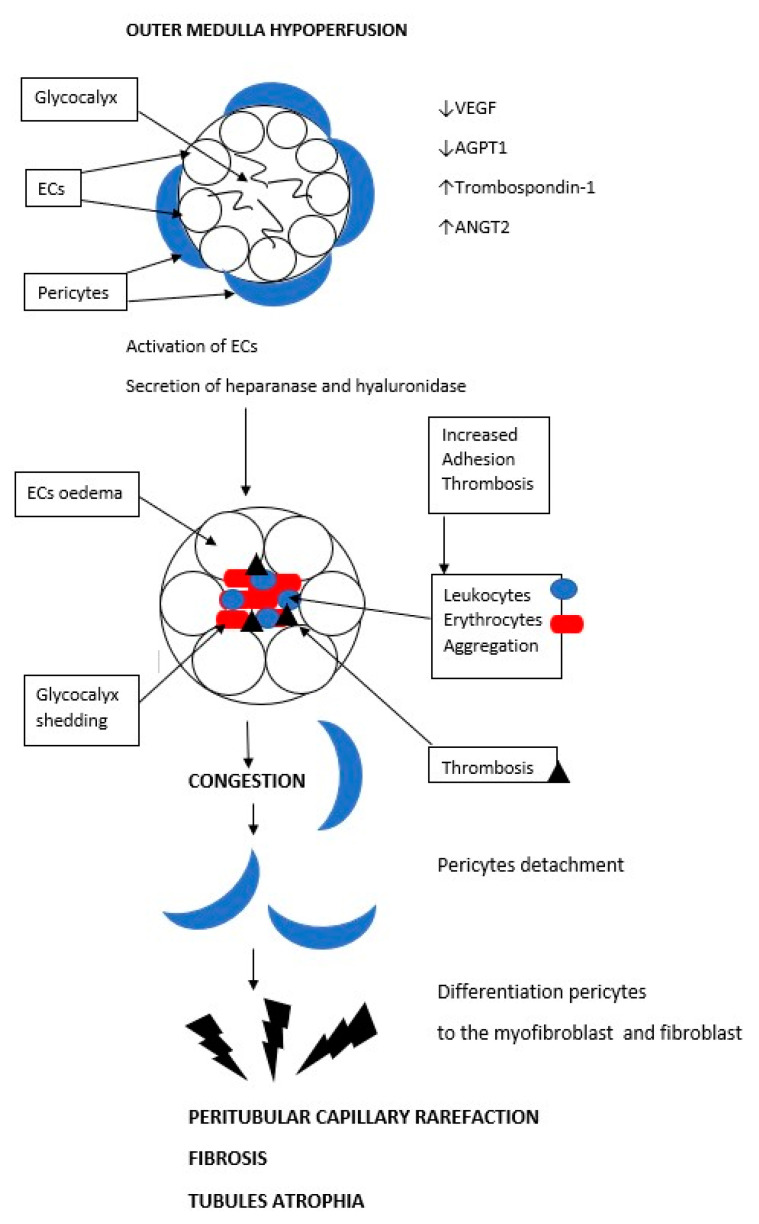
Mechanism of formation of peritubular capillary rarefaction. Consequence of acute ischemia and a cause of progression to chronic kidney disease.

## Data Availability

Not applicable.

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
