# Peer review of "Renal Microcirculation Injury as the Main Cause of Ischemic Acute Kidney Injury Development"

_biology, 2023, doi:10.3390/biology12020327_

Round 1

Reviewer 1 Report

The authors aim to describe the anatomy and function of the microcirculation, ways to prevent its damage, with particular emphasis on drugs that inhibit the renin-angiotensin-aldosterone system.

- While the topic is interesting, the article is not appropriately structured.

- There is only the “introduction” section, and some paragraphs within this section.

- It is not clear the literature search strategy.

- Overall considered the article is very poorly written, and in this format it’s too short, it seems like a long abstract.

- There are several aspects that were not considered and discussed along the article.

Therefore, I do not support for publication this paper in the current format.

Author Response

Answer to reviewer 1

Dear reviewer 1

Thank you very much for critical review of our article “ Renal microcirculation injury as the main cause of acute kidney injury development. The roles of drugs inhibiting the renin–angiotensin–aldosterone system.” We have carefully considered your suggestions and have revised our manuscript accordingly; we hope that these changes meet with your approval.

Reviewer 2 Report

Reviewing « Renal microcirculation injury as the main cause of acute kidney 2 injury development. The roles of drugs inhibiting the renin–3 angiotensin–aldosterone system » The authors give a narrative review on the relations between microvascular anatomy in kidney and microcirculatory dysfunction in acute kidney injuries. All the parts about anatomy and pathophysiology are well constructed, I can be more critic on the therapeutical parts. Half of the references are missing in the manuscript. Major comments: In the abstract: - The lecture of the abstract is not fluid, with general considerations and then details on some scattered points. In the outer medulla of the kidney part: - « which releases vasopressin that constricts the afferent arteriole… »this is wrong, the macula densa secrets renin the renin angiotensin system is then systemically involved and this is angiotensin II that acts as a major vasoconstrictor of the efferent arteriole. - Reference number 14, this is not about ischemia, this is a subtotal nephrectomy model, not the same mechanisms, this quotation is off topic. In the microcirculation part: - Give references, between lines 229 and 236. - Provide more details on the effects of each phenomenon “Microcirculatory ischemia causes damages… microcirculatory failure and increased hypoxia” lines 229-234. - “We can increase filtration, but only when there is a medullary flow.” I don’t understand the purpose of this sentence, is this a therapeutic goal? a hypothetical ideal phenomenon that should happen to decrease AKI lesions? - Line 249, please mention team works, it looks like the first author worked alone. The quotation of only one author appears many times in the text. - The complex relations between macro and microcirculation are underlined in this section, this is sure that a strong literature focuses on this point. Stronger articles could be cited in this section 10.1056/NEJMra064398. In the therapy part: 2 - There is confusion between use of RAAS blockers at the time of AKI and during hypoperfusion on one hand (where RAAS blockers are most of the time stopped because they basically drop the mean arterial pressure and thus decrease renal perfusion), and RAAS blockers after AKI, to stop chronic adverse effects of AngII. It is not possible to claim that RAAS blockers are beneficial at the initial part of AKI, whereas AngII perfusion is used in patients with shock around the world, without significant increase in AKI in large trials 10.1056/NEJMoa1704154 (supplementary materials). - Anyway, growing medical evidence suggests that RAAS blockers may be useful especially for cardiovascular complications after the acute phase of AKI, I totally agree with this, but you should separate clearly these two time points. - Some studies found more severe kidney outcomes at the acute phase in patients treated with RAAS blockers, but no worsening in longer outcomes 10.1016/j.jcrc.2022.153986. In the references: - References from 51 to the last re missing. Minor comments: In the abstract: In the introduction: - This sentence is futile « It might seem 36 that kidneys should not be susceptible to ischemia, but in reality they are easily damaged 37 by this condition. » In the renal vascularisation part: - Please give references about the renal vascularisation. In the outer medulla of the kidney part: - Be careful with too long sentences. - The refence number 14 should be at the end of the sentence. - The reference number 14 has been published in 2008, not in 2018. - “Any other 118 changes were a consequence of this disorder. » This is difficult to prove, observing successive phenomenon do not mean that they are mechanistically linked. - Reference 15 « in animals » please be more precise. - Lines 144 and 145, specify that it is seen in experimental models, things are probably getting more complex in human diverse forms of AKI. In the microcirculation part: 3 - Please give a reference about the A-V fistulae (lines 226-229). - Line 226 “animal modeL” the L is missing. In the peritubular capillaries part: - Line 290 a “.” Should be removed between text and reference. - Line 303, maybe a word is missing, I don’t understand the sentence; “proliferation potential” maybe? - Line 314 t4 instead of [74]. - The ideas of this section could be better organised, with a conclusion sentence. In the therapy part: - Line 326, “thr system”? - Lines 327 to 332 are redundant - The bottleneck effect and the use of vasodilatory agents to improve general microcirculation in shock could be mentioned as a therapeutic perspective 10.1186/s13613-019-0577-9

Author Response

Answer to reviewer 2.

Dear reviewer 2.

Thank you very much for critical review of our article “ Renal microcirculation injury as the main cause of acute kidney injury development. The roles of drugs inhibiting the renin–angiotensin–aldosterone system.” We have carefully considered your suggestions and have revised our manuscript accordingly; we hope that these changes meet with your approval.

Renal microcirculation injury as the main cause of acute kidney 2 injury development. The roles of drugs inhibiting the renin–3 angiotensin–aldosterone system » The authors give a narrative review on the relations between microvascular anatomy in kidney and microcirculatory dysfunction in acute kidney injuries. All the parts about anatomy and pathophysiology are well constructed, I can be more critic on the therapeutical parts.

  1. Half of the references are missing in the manuscript. I filled in the missing references.
  2. Major comments: In the abstract: - The lecture of the abstract is not fluid, with general considerations and then details on some scattered points. I changed the content of the abstract
  3. In the outer medulla of the kidney part: - « which releases vasopressin that constricts the afferent arteriole… »this is wrong, the macula densa secrets renin the renin angiotensin system is then systemically involved and this is angiotensin II that acts as a major vasoconstrictor of the efferent arteriole. Tubuloglomerular feedback is connected with activation macula densa by high concentration ofNaCl which constrict afferent arteriole. This reaction is connceted with secretion of ATP/adenozyn and paracrinely vasopresin. I changed citation and add inforamtion about ATP/adenosin. High concentration of NaCl in macula densa inhibit renin exretion.

  1. - Reference number 14, this is not about ischemia, this is a subtotal nephrectomy model, not the same mechanisms, this quotation is off topic. I changed citation.

  1. In the microcirculation part: - Give references, between lines 229 and 236. - Provide more details on the effects of each phenomenon “Microcirculatory ischemia causes damages… microcirculatory failure and increased hypoxia” lines 229-234. – I made changes.
  2. “We can increase filtration, but only when there is a medullary flow.” I don’t understand the purpose of this sentence, is this a therapeutic goal? a hypothetical ideal phenomenon that should happen to decrease AKI lesions? – I changed sentence.
  3. Line 249, please mention team works, it looks like the first author worked alone. The quotation of only one author appears many times in the text. I made changes.
  4. The complex relations between macro and microcirculation are underlined in this section, this is sure that a strong literature focuses on this point. Stronger articles could be cited in this section 10.1056/NEJMra064398. I have attached this article.
  5. In the therapy part: 2 - There is confusion between use of RAAS blockers at the time of AKI and during hypoperfusion on one hand (where RAAS blockers are most of the time stopped because they basically drop the mean arterial pressure and thus decrease renal perfusion), and RAAS blockers after AKI, to stop chronic adverse effects of AngII. It is not possible to claim that RAAS blockers are beneficial at the initial part of AKI, whereas AngII perfusion is used in patients with shock around the world, without significant increase in AKI in large trials 10.1056/NEJMoa1704154 (supplementary materials). - Anyway, growing medical evidence suggests that RAAS blockers may be useful especially for cardiovascular complications after the acute phase of AKI, I totally agree with this, but you should separate clearly these two time points. - Some studies found more severe kidney outcomes at the acute phase in patients treated with RAAS blockers, but no worsening in longer outcomes 10.1016/j.jcrc.2022.153986. I changed my sentences recommending the inclusion of RAAS inhibitors after blood pressure and volume restoration. I cited a large meta-analysis confirming the benefit of such a procedurÄ™.
  6. In the references: - References from 51 to the last re missing. I filled in the missing references
  7. Minor comments: In the abstract: In the introduction: - This sentence is futile « It might seem 36 that kidneys should not be susceptible to ischemia, but in reality they are easily damaged 37 by this condition. I changed abstract.
  8. » In the renal vascularisation part: - Please give references about the renal vascularisation. I gave references.
  9. In the outer medulla of the kidney part: - Be careful with too long sentences. - The refence number 14 should be at the end of the sentence. - The reference number 14 has been published in 2008, not in 2018. Any other 118 changes were a consequence of this disorder. I made changes.
  10. - Reference 15 « in animals » please be more precise. I made changes.
  11. - Lines 144 and 145, specify that it is seen in experimental models, things are probably getting more complex in human diverse forms of AKI. I made changes.
  12. In the microcirculation part: 3 - Please give a reference about the A-V fistulae (lines 226-229). I added the references.-
  13. Line 226 “animal modeL” the L is missing. I made changes.
  14. In the peritubular capillaries part: - Line 290 a “.” Should be removed between text and reference. – I made changes.
  15. Line 303, maybe a word is missing, I don’t understand the sentence; “proliferation potential” maybe? - I made changes.
  16. Line 314 t4 instead of [74]. - The ideas of this section could be better organized, with a conclusion sentence. I added conclusion.
  17. In the therapy part: - Line 326, “thr system”? - Lines 327 to 332 are redundant - I deleted those sentences
  18. The bottleneck effect and the use of vasodilatory agents to improve general microcirculation in shock could be mentioned as a therapeutic perspective 10.1186/s13613-019-0577-9 I mentioned about this phenomenon, and cited this article.

Reviewer 3 Report

The manuscript by Kwiatkowska et al. reviews recent papers describing the transition from acute kidney injury to chronic renal disease. This topic is interesting and attracts much attention in kidney field. The authors provide insightful thoughts about the the role of renal microcirculation and strengthen the importance of renin-angiotensin-aldosterone system. The review should be of interest to relevant scientific community. Before it is accepted one point needs to be clarified. The glomerulus or ball of capillaries should be indicated in the Figure 1 legend.

Author Response

Answer to reviewer 3

Dear Reviewer 3,

Thank you very much for critical review of our article “ Renal microcirculation injury as the main cause of acute kidney injury development. The roles of drugs inhibiting the renin–angiotensin–aldosterone system.” We have carefully considered your suggestions and have revised our manuscript accordingly; we hope that these changes meet with your approval.

The manuscript by Kwiatkowska et al. reviews recent papers describing the transition from acute kidney injury to chronic renal disease. This topic is interesting and attracts much attention in kidney field. The authors provide insightful thoughts about the the role of renal microcirculation and strengthen the importance of renin-angiotensin-aldosterone system. The review should be of interest to relevant scientific community. Before it is accepted one point needs to be clarified. The glomerulus or ball of capillaries should be indicated in the Figure 1 legend.

I changed the Figure 1.

Round 2

Reviewer 1 Report

Unfortunately, the new version did not adequately address my previous comments, and in particular:

- There is only the “introduction” section, and some paragraphs within this section.

- It is not clear the literature search strategy.

- Overall considered the article is very poorly written, and in this format it’s too short, it seems like a long abstract.

Actually, the structure of the article as well as the information did not change.

Author Response

Answer to reviewer 1

Dear reviewer 1

Thank you very much for critical review of our article “ Renal microcirculation injury as the main cause of acute kidney injury development. The roles of drugs inhibiting the renin–angiotensin–aldosterone system.” We have carefully considered your suggestions and have revised our manuscript accordingly; we hope that these changes meet with your approval.

We have changed the layout of the article and the numbering of sections, we rewrote abstract and introduction, corrected grammatical errors, we have attached a second schematic drawing and change title.

Ewa Kwiatkowska
